**Calibration and analysis of the uncertainty in downscaling global land use and land cover projections from GCAM using Demeter (v1.0.0)**

Min Chen[1*], Chris R. Vernon[2], Maoyi Huang[2], Katherine V. Calvin[1], and Ian P. Kraucunas[2]

[1] Joint Global Change Research Institute, Pacific Northwest National Laboratory, College Park, Maryland 20740, United States

[2] Atmospheric Sciences and Global Change Division, Pacific Northwest National Laboratory, P.O. Box 999, Richland, Washington 99352, United States

[*]Corresponding author

Email: min.chen@pnnl.gov

Telephone: 1-301-314-6755

Fax: 1-301-314-6719

**Abstract**

Demeter is a community spatial downscaling model that disaggregates land use and land cover
changes projected by integrated human-Earth system models. Demeter has not been intensively
calibrated, and we still lack a good knowledge about its sensitivity to key parameters and the parameter
uncertainties. We used long-term global satellite-based land cover records to calibrate key Demeter
parameters. The results identified the optimal parameter values and showed that the parameterization
substantially improved the model's performance. The parameters of intensification ratio and selection
threshold were the most sensitive and needed to be carefully tuned, especially for regional applications.
Further, small parameter uncertainties after calibration can be inflated when propagated into future
scenarios, suggesting that users should consider the parameterization equifinality to better account for the
uncertainties in the Demeter downscaled products. Our study provides a key reference for Demeter users,
and ultimately contribute to reducing the uncertainties in Earth system model simulations.

**Key words**: Demeter; land use and land cover change; parameterization; human-Earth systems models

## 1. Introduction

Land Use and Land Cover Change (LULCC) represents one of the most important human impacts on the Earth system (Hibbard et al., 2017). Besides its socioeconomic effects, LULCC is directly linked to many natural land surface processes, such as land surface energy balance, carbon and water cycle (e.g., Piao *et al* 2007, Law *et al* 2018, Sleeter *et al* 2018, Pongratz *et al* 2006), and indirectly affects the climate system (e.g., Dickinson and Kennedy 1992, Findell *et al* 2017, Costa and Foley 2000). Thus, LULCC has been considered as a key process in simulating of Earth system dynamics, and LULCC inputs at appropriate time steps and spatial resolutions are required to match the setup of the Earth System Models (ESMs) and the nature of spatial heterogeneity of the Earth system processes (Brovkin et al., 2013; Lawrence et al., 2016; Prestele et al., 2017).

While recent historical LULCC information can be obtained by ground investigation or satellite remote sensing (Friedl et al., 2002; Hansen et al., 2000; Loveland et al., 2000; Zhang et al., 2003), projections of future LULCC largely rely on mathematical models that bring socioeconomic and other diverse sectoral information together in a coherent framework to simulate the interactions between natural and human systems. However, these integrated models project LULCC at subregional level, i.e., the basic spatial units that have uniform properties for every sector (e.g., agricultural, energy and water etc.), typically ranging from a few hundred to millions of square kilometers (Edmonds et al., 2012). For example, the GCAM model has been widely used to explore future societal and environmental scenarios under different climate mitigation policies which provides LULCC projections at region-agroecological or water basin level (Edmonds et al., 1997; Edmonds and Reilly, 1985; Kim et al., 2006). ESMs divide the Earth surface into a number of grid cells and the forcing data have to be available at the same spatial resolution to drive the ESMs (Taylor et al., 2012). Therefore, spatial downscaling of the subregional LULCC becomes a critical step for linking models like GCAM and ESMs to investigate the effects of the LULCC on the processes in the natural world, and further the interactions between the human and natural systems (Hibbard and Janetos, 2013; Lawrence et al., 2012).

There has been a few spatial disaggregation studies for LULCC, e.g., the Global Land Use Model (Hurtt et al., 2011) and a dynamic global land use model (Meiyappan et al., 2014) with various geographical and socioeconomic assumptions. In previous studies, we have developed a new simple and efficient LULCC downscaling model, named Demeter (version 1.0.0), to bridge GCAM and ESMs (Le Page et al., 2016; Vernon et al., 2018; West et al., 2014), and made it available online at http://doi.org/10.5281/zenodo.1214342 . Comparing to other models, Demeter makes minimal assumptions of the socioeconomic impacts. Instead, it uses a few parameters to implicitly characterize the spatial patterns of land use changes (See introductions in Section 2.1). Demeter has been successfully applied at both global (Le Page et al., 2016) and regional (West et al., 2014) levels for downscaling GCAM-projected land use and land cover changes, and has been further developed with an extensible

output module which streamlines producing specific output formats required by various ESMs (Vernon et
al., 2018).  However, Demeter's parameters (discussed in Section 2.1), which conclude many geographic
patterns of long-term land cover changes such as intensification and expansion, are difficult to determine
by either literature review or simple mathematical calculations. Therefore, Demeter's parameter values
were empirically determined and a complete analysis on Demeter's parametric sensitivity and
uncertainties as well as a rigorous model calibration has not been conducted to help minimize the
propagation of downscaling errors. In recent years, a growing number of long-term global remote-
sensing-based LULCC datasets are made available (e.g., the Land Cover project of the European Space
Agency Climate Change Initiative, MODIS Land Cover product collections 6), it becomes possible to use
these datasets to calibrate Demeter parameters. The major objective of this study is to develop a
framework for calibrating the key parameters of Demeter, testing and quantifying the parameter
sensitivities and uncertainties, and demonstrating how the parameter uncertainties would affect
downscaled products.

**2.  Method**
*2.1 Demeter*
Demeter is a land use and land cover change downscaling model, which is designed to disaggregate
projections of land allocations generated by GCAM and other models. For example, GCAM projects land
cover areas in each of its spatial units (e.g., region-agro-ecological zones, region-AEZ) for each land
cover type, and Demeter uses gridded observational land cover data (e.g., satellite-based land cover
product) as the reference spatial distribution of land cover types and allocates the GCAM-projected land
area changes to grid level at a target spatial resolution, following some user-defined rules and spatial
constraints (Figure S1). Below we briefly summarize the key processes of Demeter, and the detailed
algorithms can be found in three earlier publications (Le Page et al., 2016; Vernon et al., 2018; West et
al., 2014).
Demeter first reconciles the land cover classes defined in the parent model and the reference dataset
to user-defined unified final land types (FLTs). Downscaled land cover types will be presented in FLTs.
For example, if Demeter reclassifies the 22 GCAM land cover types and the 16 International Geosphere-
Biosphere Programme (IGBP) land cover types from the reference dataset into 7 FLTs (Forest, Shrub,
Grass, Crops, Urban and Sparse), the 7 FLTs will be the land types represented in Demeter's outputs by
default. Demeter then harmonizes the GCAM-projected land cover areas and the reference dataset at the
first time step (or 'base year') to make sure they are consistent with the GCAM spatial units and allocates
the projected land cover changes by intensification and extensification. Intensification is the process of
increasing a particular land cover in a grid cell where it already exists, while extensification creates new
land cover in grid cells where it does not yet exist but is in proximity to an existing allocation. The order
of transitions among land cover types is defined by "transition priorities" during the processes of
intensification and extensification. A parameter ($r$, from 0 to 1) is defined as the ratio of intensification,
and thus 1-$r$ of the land cover change is for extensification. Proximal relationships are defined by spatial
constraints that determine the probability that a grid cell may contain a particular land use or land cover
class. The current Demeter setup includes three spatial constraints: kernel density (KD), soil workability
(SW) and nutrient availability (NA). KD measures the probability density of a land cover type around a
given grid cell, and SW and NA are normalized scalars (0~1) for agricultural suitability.  For each land
cover type and grid cell, KD is calculated by the spatial distance ($D$) at the runtime, and SW and NA are
estimated from the Harmonized World Soil Database (HWSD, FAO/IIASA/ISRIC/ISSCAS/JRC, 2012).
A suitability index ($SI$) from 0 to 1 is defined as the weighted-average of the three spatial constraints to
assess how suitable a grid cell is to receive a land cover type:

$$SI = (w_K*KD+w_S*SW+w_N*NA)/(w_K+w_S+w_N) \tag{1}$$

where $w_K$, $w_S$, and $w_N$ are the weights for KD, SW and NA, respectively, and the sum of them is 1. In the
process of extensification, Demeter ranks candidate grid cells based on their suitability indices and selects
the most suitable candidate grid cells following a user-defined threshold percentage ($\tau$) for extensification.
In other words, $\tau$ determines the number of grid cells to be selected and used for the tentative and actual
conversion of land cover types.

Table 1. Transition priorities by analyzing the 24-year global land cover records from the Land Cover
CCI project of the European Space Agency Climate Change Initiative. The rows and columns represent
the origins and destinations of the transitions, respectively. The smaller numbers indicate higher transition
priorities.

| Final Land Types (origins) | Final Land Types (destinations) | | | | | | |
|---|---|---|---|---|---|---|---|
| | Forest | Shrub | Grass | Crop | Urban | Snow | Sparse |
| Forest | 0 | 2 | 3 | 1 | 4 | 5 | 6 |
| Shrub | 2 | 0 | 3 | 1 | 4 | 5 | 6 |
| Grass | 1 | 2 | 0 | 3 | 5 | 6 | 4 |
| Crop | 2 | 3 | 1 | 0 | 5 | 6 | 4 |
| Urban | 1 | 4 | 3 | 2 | 0 | 6 | 5 |
| Snow | 2 | 3 | 4 | 1 | 5 | 0 | 6 |
| Sparse | 2 | 3 | 4 | 1 | 5 | 6 | 0 |


*2.2  Calibrate Demeter with historical land cover record and sensitivity analysis*

124       As indicated above, users should define a few parameters including the treatment order, the transition

priorities for allocating the land cover changes, the intensification ratio $r$, the selection threshold $\tau$, the
radius for calculating kernel density $D$, and weights for the spatial constraints ($w_K$, $w_S$, and $w_N$), in order to
use Demeter for downscaling projected land cover change. These parameters were determined empirically
in previous studies. Here we calibrated these parameters for Demeter using a time series of global land
cover records from the Land Cover project of the European Space Agency Climate Change Initiative
(referred to as CCI-LC products hereafter). The CCI-LC products have been generated by critically
revisiting all algorithms required for the generation of a global land cover product from various Earth
Observation (EO) instruments, thus provide a globally consistent land cover record over two decades
(1992-2015). The CCI-LC products are available at 300 m spatial-resolution and annual time step and
classify the global land cover into 38 groups. We reclassified the CCI-LC products into the default 7
FLTs (Table S1) and resampled them into 0.25° resolution with the official software tools, following the
description of CCI-LC products in the user guide
(http://maps.elie.ucl.ac.be/CCI/viewer/download/ESACCI-LC-Ph2-PUGv2_2.0.pdf). Figure 1 shows
large interannual global changes for the 7 FLT areas, especially for the forests and croplands, which have
decreased and increased over 0.6 million $km^2$ over the past two decades, respectively. We used the
gridded 0.25° CCI-LC over the 24-year period as the observational data (below referred to "LC-grid-
obs") and aggregated them into GCAM's region-AEZ level to produce a synthetic GCAM-projected land
cover change (below referred to "LC-AEZ-syn"). In this way, we can apply Demeter to LC-AEZ-syn to
calibrate Demeter with the LC-grid-obs by tuning the parameters of Demeter.

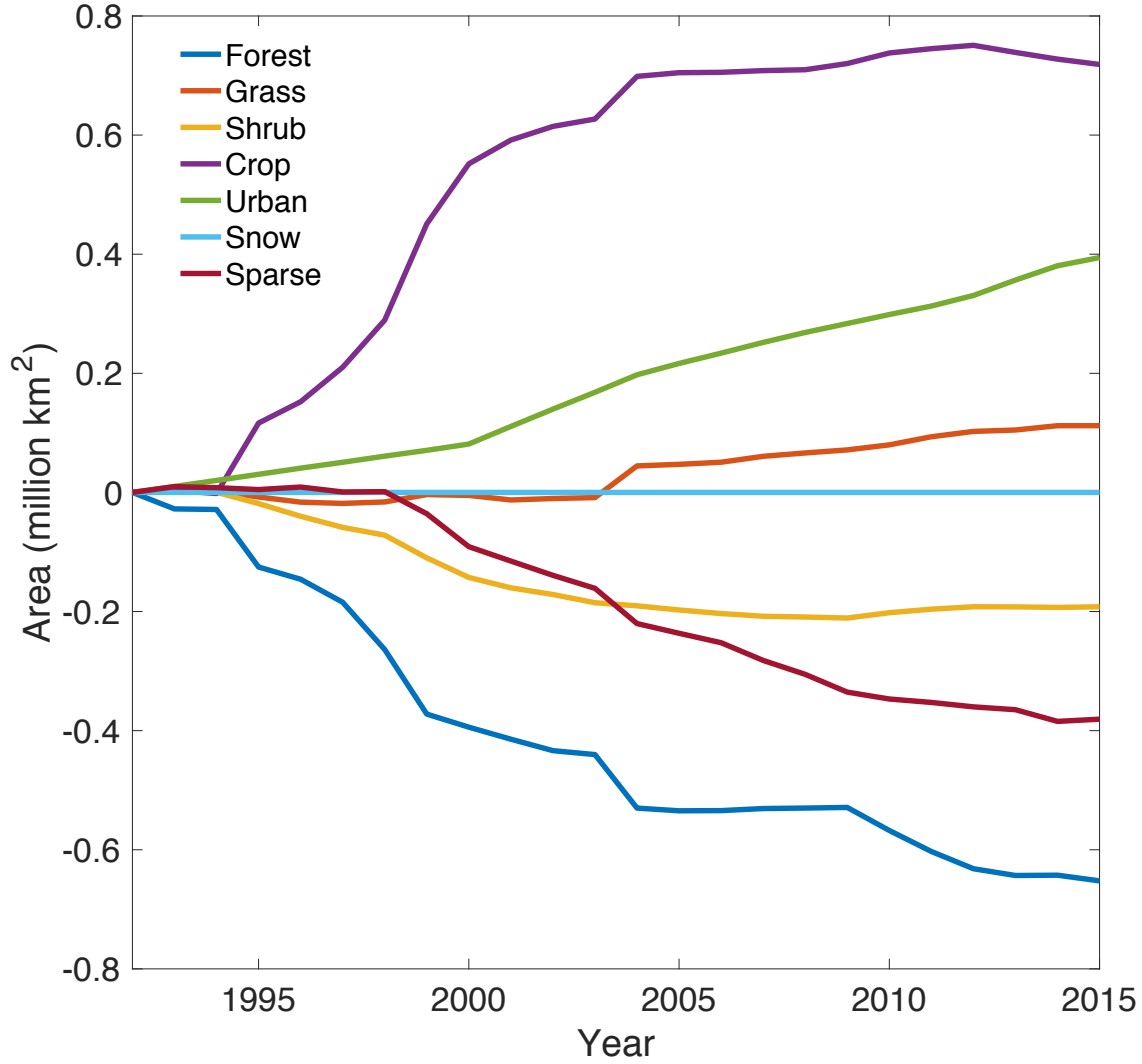


Figure 1. Interannual changes of global Final Land Types (FLTs) areas over 1992-2015 relative
to 1992, as indicated by the ESA CCI-LC product.

148       A preliminary sensitivity analysis of Demeter indicated that the downscaled results are not sensitive

to treatment order and transition priorities (Le Page et al., 2016), thus we used the default treatment order,
i.e., from least to greatest: Urban, Snow, Sparse, Crops, Forest, Grass, Shrub. We decided the transition
priorities by sorting the probabilities of transitioning one FLT to another based on the 24-year CCI-LC
record (Table 1). To calibrate the other six parameters ($r$, $\tau$, $w_K$, $w_S$, $w_N$ and $D$), we sampled their values at
equal intervals (Table 2) and generated all possible combination (23,100 in total) for a Monte-Carlo
ensemble Demeter downscaling experiment, using LC-AEZ-syn as the input. The Monte-Carlo
experiment generated 23,100 sets of downscaled 0.25-degree global land use and land cover areas, which
were compared against LC-grid-obs to calculate their similarities to the observational data, ranked by
their discrepancies from the least to greatest to determine the likelihood of the parameters. We calculated
the discrepancies as the root mean square error ($E_y$) between the downscaled and observed land cover
areas for each year:

$$E_y = \sqrt{\frac{1}{G}\frac{1}{L}\sum_{g}^{G}\sum_{l}^{L}\left(Ad_{y,l,g} - Ad_{o,l,g}\right)^2}$$

160                                                                                                      (2)

and the average of the discrepancies over the years ($E$):

$$E = \frac{1}{Y}\sum_{y}^{Y}E_y$$

162                                                                                                      (3)

where $g$ is the index for $G$ grid cells over the globe ($G = 265{,}852$), $l$ is the index for the $L$ FLTs ($L = 8$), $y$
is the index for $Y$ years. We chose 1992, 2000, 2005, 2010 and 2015 to keep consistent with the GCAM
time steps, thus $Y = 5$. $Ad_{y,l,g}$ and $Ao_{y,l,g}$ are the downscaled and observational land cover areas for grid
cell $g$, FLT $l$ and year $y$. The unit for $E_y$ and $E$ is km$^2$.
To test the model sensitivity to these key parameters, we conducted a sensitivity analysis using the
results from the Monte-Carlo experiment. The first-order and total-order Sobol sensitivity indices were
used to identify the model sensitivity to each of the six parameters (Saltelli et al., 2004). Let $\theta_i$ denotes the
$i$th parameter ($i=1,\ldots,n$, here $n=6$), $\varepsilon$ is the model outputs (i.e., the discrepancies between downscaled and
observed land cover areas), the first-order Sobol index ($S_i$) is defined as:

$$S_i = \frac{Var\left[E\left(\varepsilon\,|\,\theta_i\right)\right]}{Var\left(\varepsilon\right)}$$

172                                                                                                      (4)

Here $Var$ and $E$ are the statistical variance and expectation. And the total-order Sobol index ($S_{Ti}$) is
defined as the sum of sensitivity indices at any order involving parameter $\theta_i$, where $S_{ijk\ldots n}$ denotes the $n$th-
order sensitivity index:

$$S_{Ti} = S_i + \sum_{j=1,j\neq i}^{n} S_{ij} + \sum_{j,k=1,j,k\neq i}^{n} S_{ijk} + \ldots + \sum_{j,k\ldots n=1,j,k,\ldots,n\neq i}^{n} S_{ijk\ldots n}$$

176                                                                                                      (5)

The first-order Sobol index represents the contribution to the output variance of the main effect of $\theta_i$,
therefore it measures the effect of varying $\theta_i$ alone; and the total-order Sobol index measures the
contribution to output variance of $\theta_i$ and includes all variance caused by its interactions with other
parameters. Larger Sobol indices indicate higher parameter sensitivities.



Table 2. Key parameters, and their sampling range and steps for calibration in this study.

| Name | Definition | Min | Max | Sampling step |
|---|---|---|---|---|
| $w_N$ | Weight of soil nutrient availability for calculating suitability index | 0 | 1 | 0.2 |
| $w_S$ | Weight of soil workability for calculating suitability index | 0 | 1 | 0.2 |
| $w_K$ | Weight of kernel density for calculating suitability index | 0 | 1 | 0.2 |
| $r$ | Intensification ratio | 0 | 1 | 0.1 |
| $\tau$ | Selection threshold | 0 | 1 | 0.1 |
| $D$ | Kernel radius | 10 | 100 | 10 |


*2.3 Propagate the parameter uncertainties to GCAM LULCC downscaling*
We selected parameter combinations which produced the smallest 5% *E*s based on their rankings
from the Monte-Carlo experiment, and used them as 'acceptable' parameters to represent the parameter
uncertainties after calibration. We used Demeter with these parameters to downscale the GCAM-
projected LULCC at 5-year time step from 2005 to 2100 under a reference scenario to examine the
uncertainties of land cover areas for each FLT to demonstrate how different the downscaled LULCC can
be induced by the uncertain parameters. The reference scenario is a business-as-usual case with no
explicit climate mitigation efforts that reaches a higher radiative forcing level of over 7 W m$^{-2}$ in 2100.
We only saved the downscaling results in 2005, 2010, 2050 and 2100 considering the size of the output
files and computational cost. Finally, we calculated the standard deviation across the downscaled land
cover areas for each FLT driven by different parameter combinations, which indicates the parameter-
induced model uncertainties.

**3. Results**
*3.1 Parameter estimation and sensitivity*
The Monte-Carlo Demeter experiment driven by the 23,100 ensemble parameter sets produced
diverse downscaled LULCC realizations. As shown in Figure 2a, the disagreements between the
downscaled FLT fraction and the reference record, measured by the average root mean square error (*E*,
Equation 3) for all the FLTs and grid cells over the five years (1992, 2000, 2005, 2010 and 2015), are
mainly distributed between 8 and 17 km$^2$ (about 1%-3% of the area of a 0.25-degree grid cell).

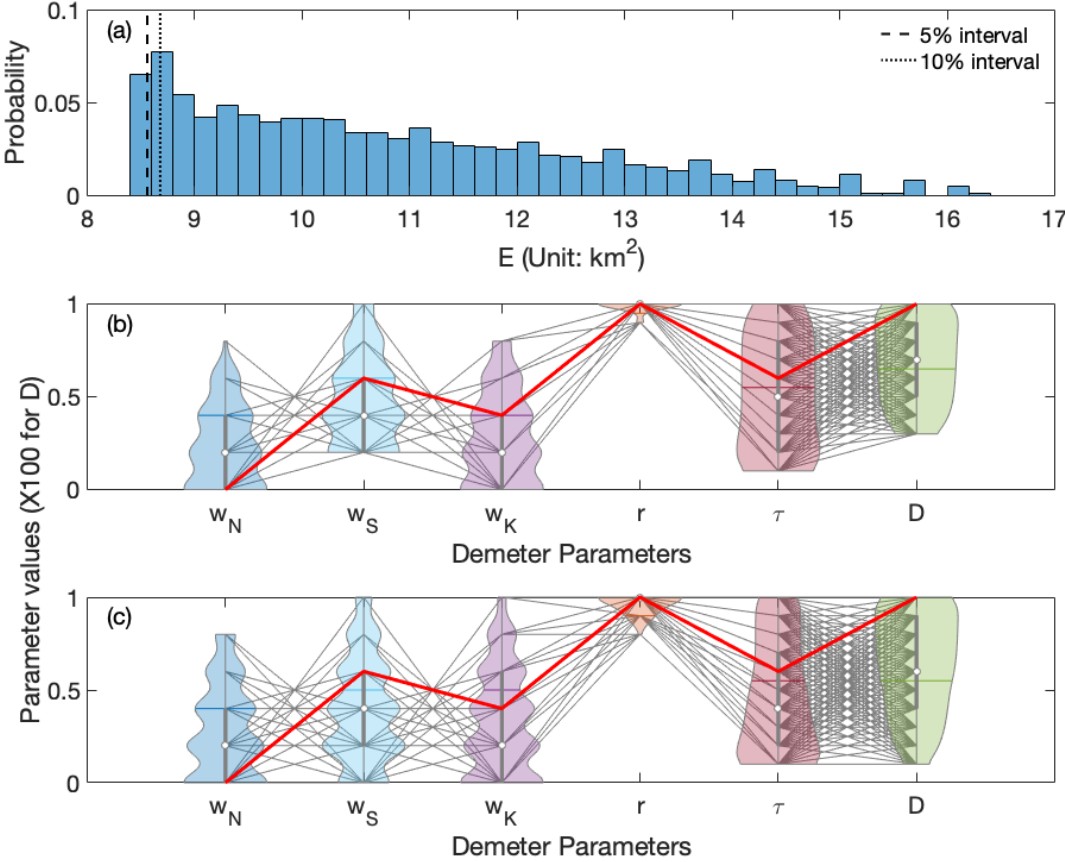


Figure 2. (a) Histogram of the $E$s, i.e., the global average discrepancies between the downscaled
and observed land cover areas with the 23,100 ensemble parameter sets; the vertical dashed line
in (a) shows the interval of the 'acceptable' 5% parameters, as described in Section 2.3; (b) the
probability density of each of the 'acceptable' 5% parameters, as shown by the violin plots; the
black lines across the six parameters show all the 'acceptable' 5% parameter sets, and the red
line indicates the global optimal parameter values; the box plots and horizontal bar inside the
violin plots indicate the interquartile ranges and the mean of the parameter values, respectively.
(c) same as (b) but shows the 'best' 10% parameter sets. Note that the values of $D$ were divided
by 100 for the purpose of illustration in (b) and (c).

Figure 3 shows the relationship between the values of the six parameters and their corresponding $E$s,
resulted from the Monte-Carlo experiment. We found that the $E$s are significantly correlated to all the six
parameters ($p<0.01$). The intensification ratio ($r$) has the strongest linear correlation with the $E$s
($R^2$=0.64), followed by the selection threshold ($\tau$) ($R^2 = 0.24$). Overall, the parameters $w_K$ and $\tau$ are
positively correlated with $E$s (positive slopes of the trendlines), while $w_N$, $w_S$, $r$ and $D$ hold negative
correlations, indicating that smaller $w_K$ and $\tau$, and larger $w_N$, $w_S$, $r$ and $D$ are associated with smaller $E$s.

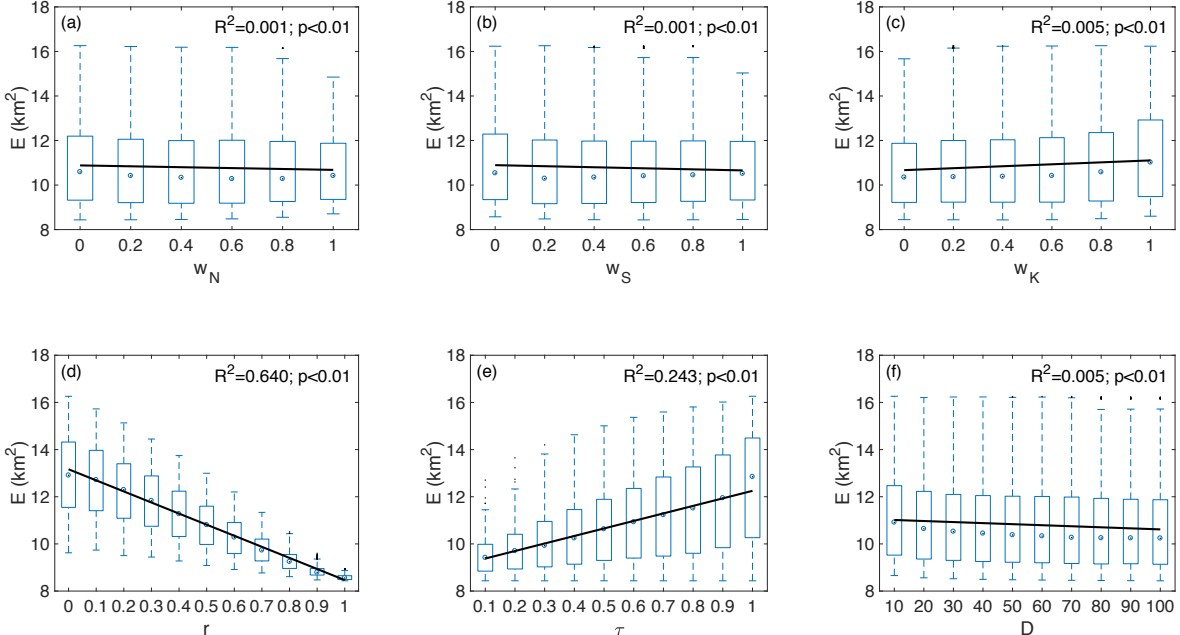


Figure 3. Relationships between the six Demeter parameters and the global average discrepancies between the downscaled and observed land cover areas ($E$s) resulted from the Monte-Carlo ensemble experiment. Box plots shows distributions of the $E$s and the solid lines show the linear trends.

Figure 4 shows the first-order and total-order Sobol indices calculated with the parameter ensemble
and the associated $E$s. As indicated by the first-order Sobol indices, the intensification ratio $r$ directly
contributes about 59% to the variability of the $E$s, followed by the selection threshold $\tau$ and kernel radius
$D$, which directly contribute 29% and 1% to the variability of the $E$s. The other parameters ($w_N$, $w_S$ and
$w_K$) have little direct contributions to the $E$ variability. The total-order Sobol indices showed similar order
of parameter importance. r and its interactions with other parameters contributed about 70% of the $E$
variability, $\tau$ contributed about 40%, $D$ contributed about 3%, and $w_N$, $w_S$ and $w_K$ contributed 2%
respectively. It is clear that the downscaling error is most sensitive to the intensification ratio, followed by
the selection threshold, but not sensitive to the kernel radius and the weighting factors of the spatial
constraints.

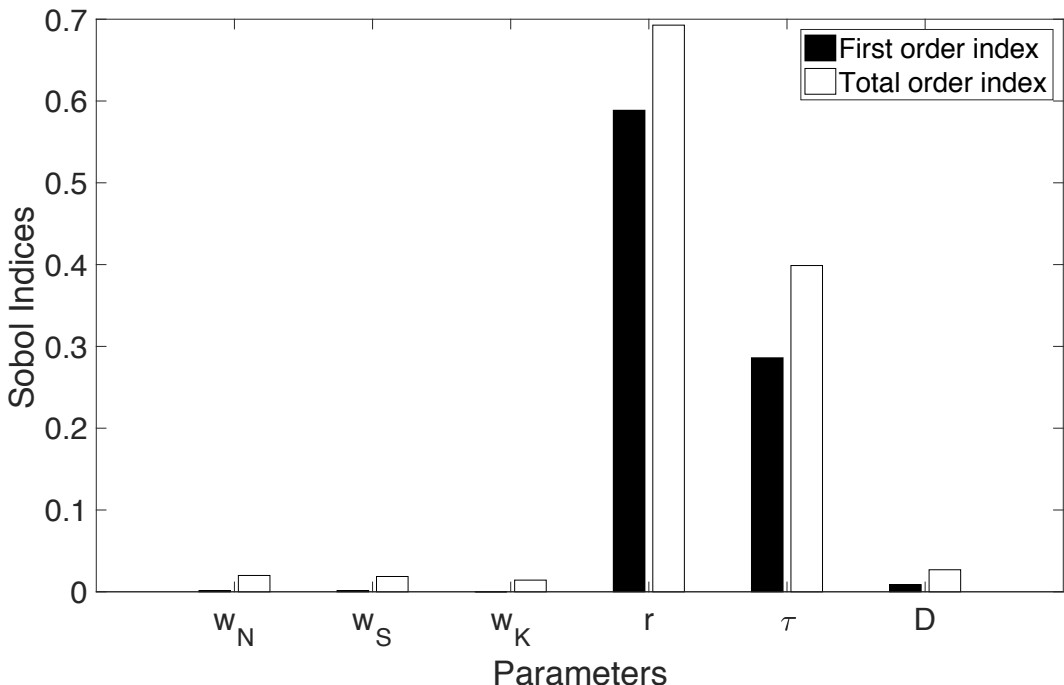


Figure 4. Sobol sensitivity indices for the six Demeter parameters. Higher indices indicate higher sensitivities.

We identified the 'best' parameters, which are associated with the lowest $E$, and marked them as the red line in Figure 2b. We also selected 'acceptable' parameters that have $E$s lower than 5% quantile in Figure 2a (hereafter referred to as 'top 5% parameters') and thus have the similar performance as the 'best' parameters (differences of E < 1%), and used them to represent the uncertainty of the parameters shown as the probability density distributions in Figure 2b. The best $w_N$, $w_S$, $w_K$, $r$, $\tau$ and $D$ are 0, 0.6, 0.4, 1, 0.6 and 100, respectively. All the parameters are constrained with the calibration comparing to their uniform prior distributions. The intensification ratio $r$ has been constrained into a small range (0.9-1.0 and mostly 1.0) from 0-1.0. Constraining on the other parameters are relatively weaker: $w_N$, $w_S$, and $w_K$ have been narrowed to the ranges of 0-0.8, 0.2-1.0, and 0-0.8, and primarily distributed in 0-0.4, 0.2-0.6 and 0-0.4 (the first and third quantiles), respectively; $\tau$ and $D$ have been constrained into the range of 0.2-1.0 and 30-100 with the first and third quantiles being 0.2-0.8 and 40-90, respectively. This analysis again indicates that $r$ is the most sensitive parameter, therefore its posterior distribution can be significantly narrowed through the calibration. In addition, we also selected the 'acceptable' parameters that have $E$s lower than 10% quantile (top 10% parameters), as shown in Figure 2a and 2c. Similar distribution of top 10% parameters are found as that of the top 5% parameters, with some small extension on the ranges of 5% parameters.


 *3.2 Performance of Demeter in downscaling LULCC*

Demeter generally performs well in downscaling the synthetic land use and land cover change with
small disagreements with the reference data. For all FLTs, the disagreements between the downscaled
FLT fraction and the reference record in 1992 (i.e., $E_{1992}$ in Equation 2), are close to zero since we used it
as the harmonization year. The disagreements in 2000 ($E_{2000}$) are mainly distributed in a range between 5
and 15 km$^2$ (about 1%-2% of a 0.25-degree grid cell), with the median about 10 km$^2$ and the mean
slightly above 12 km$^2$ (Figure 5h). The disagreements increase over years at a rate of about 1 km$^2$ per 5-
year time step and reach 13-24 km$^2$ (median: 15 km$^2$; mean: 18 km$^2$) in 2015. Overall, the average
disagreements over the five years (E) mainly distributed in 8-17 km$^2$ (also shown in Figure 2a), with the
median of about 10 km$^2$ and the mean of about 12 km$^2$.

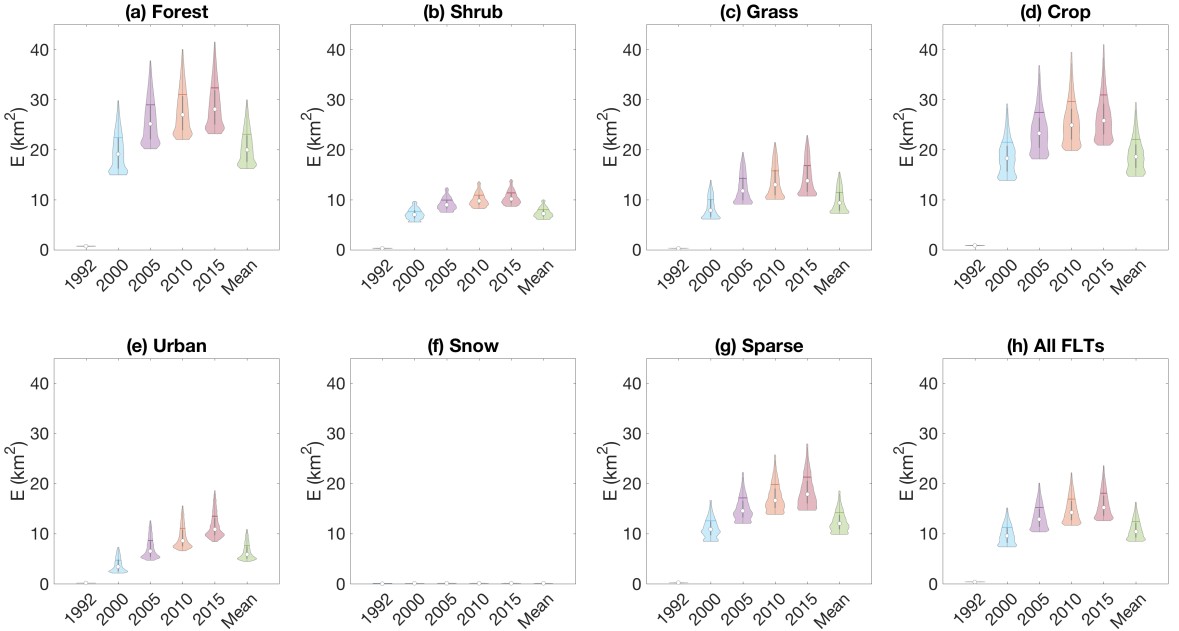


Figure 5. Possibility densities for the *E*s between downscaled and observational Final Land Type
areas for 1992, 2000, 2005, 2010, 2015 and the mean of the five time-steps. The box plots and
horizontal bar inside the violin plots indicate the interquartile ranges and the mean of the
parameter values, respectively. Note that the *E*s for Snow are close to 0 thus not visible in the
figure.

The errors for each of the FLTs follow the same increasing trend over the years. Forest and crop have
the largest disagreements between the downscaled and reference distributions with the errors are
primarily located in the range of 20-40 km$^2$ in average over the five time steps (Figure 5a,d). The errors
for sparse lands are relatively smaller, which mainly fall into the range of 10-20 km$^2$ (Figure 5g),
followed by grass, shrub and urban, with the errors are mainly distributed in 0-10 km² averagely over the
five years. Errors for snow is near zero since there was little areal change for this FLT in the CCI-LC
record (Figure 1) and little LULCC allocation was needed in the downscaling process over the years.

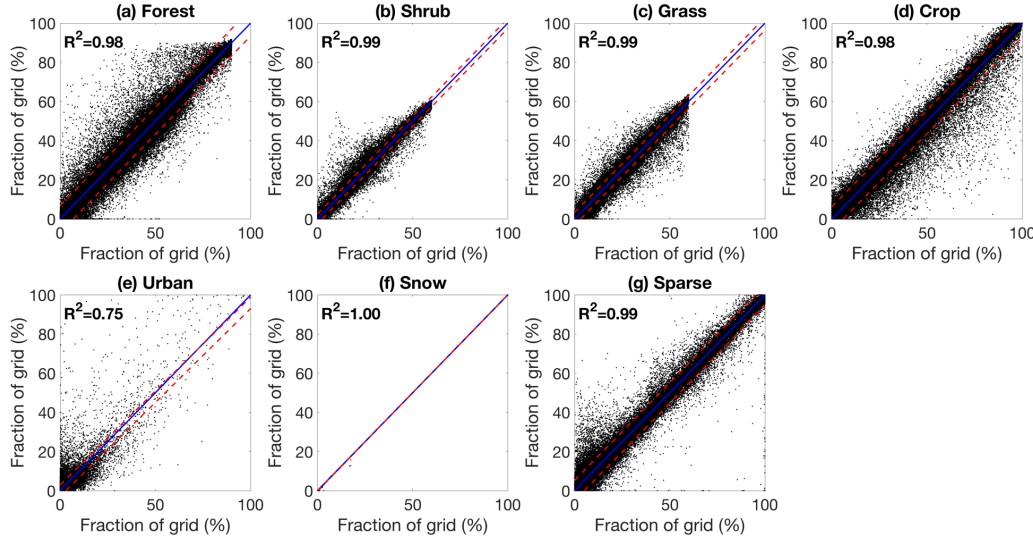


Figure 6. Comparison between the observed and downscaled Final Land Type with optimal
parameters over the 265,852 0.25-degree grid cells in 2015. The blue solid lines show the 1:1
line, and the red dashed lines show the 95% confidence intervals.
Figure 6 shows the comparison between reference gridded CCI-LC FLTs and the downscaled FLTs
driven by the best parameters (see Section 3.1) among the 265,852 0.25-degree grid cells in 2015. Except
for urban, the downscaled land cover of other FLTs match the reference record very well (all $R^2$ are above
0.98). The $R^2$ is 1 for snow due to little change of snow and ice area in the CCI-LC record. Figure 7
demonstrates the spatial distribution of FLT fraction from the reference data and best downscaled results,
together with their differences, using crop as an example. We find that the downscaled results have
successfully reproduced the spatial pattern of crops from the reference data, and similar conclusions can
be drawn for other FLTs (see Figure S2-S6; figure for Snow was not shown because of little change for
this FLT). However, misallocation of the land cover change takes places in most region-AEZs, especially
where LULCC were significant (e.g., Brazil, Eastern China, temperate Africa and Northern Euroasia;
Figure 7 and S1-S5) over the study years, likely due to the application of improper global ratio of
intensification. For example, the Northern China plain has experienced extensive urbanization by
converting a large area of cropland into urbans during the past few decades (Liu et al., 2010). However,
since the calibrated intensification ratio is high (Figure 2), Demeter tends to underestimate the urban
expansion and thus overestimate cropland area at where should be urbanized. Similarly, cropland has
been largely expanded and thus applying a high intensification ratio could not capture such changes.

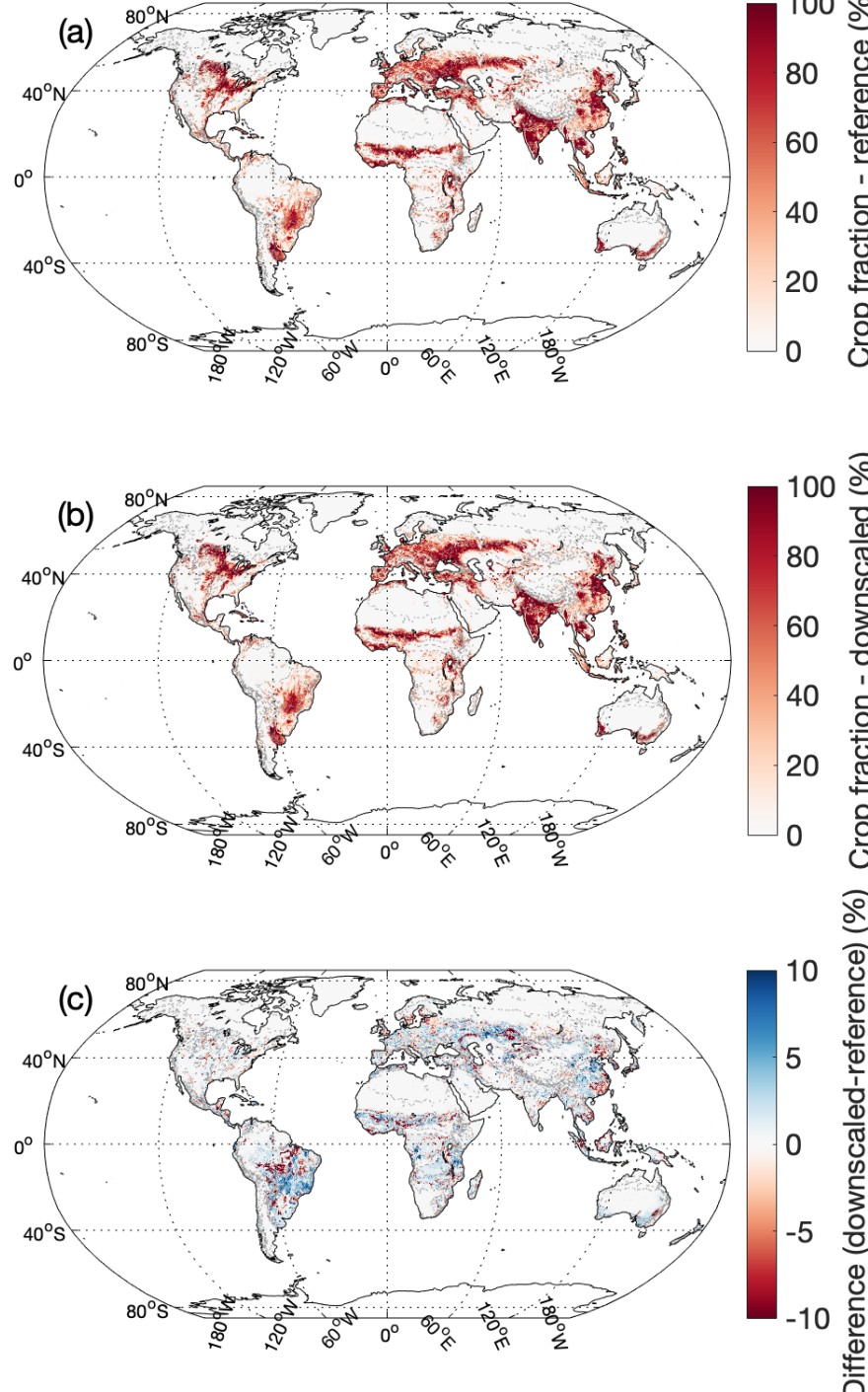


Figure 7. Spatial pattern of the observed and downscaled Crop density (measured by percentage fraction of the grid cell), and their differences in 2015. The grey dot-lines show the boundaries of the GCAM region-AEZs.

 *3.3 Uncertainty propagation*

While applying the 'acceptable' parameters (top 5% and 10%) in downscaling GCAM projections of
LULCC under the reference scenario, we found that these well-constrained parameters induced
considerable uncertainties in the downscaled results. For each grid cell, we calculated the standard
deviation ($\sigma$) of the downscaled land cover areas with different parameters for each FLT. Figure 8 shows
the mean $\sigma$ of the 265,852 0.25-degree grid cells over the globe for 2005, 2010, 2050 and 2100, as well as
the spatial variability of $\sigma$ (calculated as the standard deviation over the grid cells and shown as the
shaded area in Figure 8). As shown by the grey lines and shades in Figure 8, the uncertainty of top 5%
parameters has minor effect on downscaled Urban and Snow areas, since GCAM projected little areal
changes of urban and snow. Downscaled sparse areas were slightly affected by the choice of parameters,
indicated by small mean $\sigma$ (about 2 km$^2$ per grid cell). However, the other FLTs, including Forest, Shrub,
Grass and Crop have larger $\sigma$s, which also showed an increasing trend over time. The global mean $\sigma$ for
Forest and Shrub reached about 3 to 4 km$^2$ per grid cell and about 6 to 8 km$^2$ for Grass and Crop in 2100.
The spatial variability of $\sigma$ was also larger for these FLTs, for example, the standard deviation of $\sigma$
reached over 15 km$^2$ per grid cell in 2100 for Crop, and the maximum $\sigma$ can be over 350 km$^2$ per grid cell
in some grid cells (Figure S7). Similar results can be found by using the top 10% parameters, but with
slightly higher magnitudes (red lines and shaded areas in Figure 8 and Figure S8).

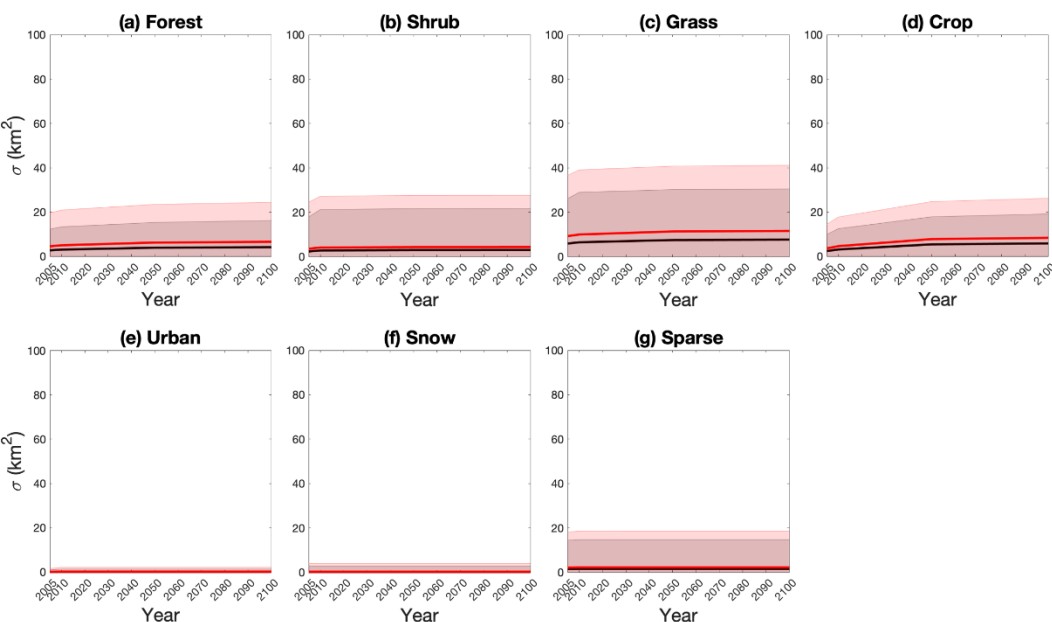


Figure 8. The Mean (shown as the solid lines) and standard deviations ($\sigma$, shown as the shaded

325         area) for the downscaled Final Land Type (FLT) areas, when propagating the parameter

uncertainties into the GCAM-projected land use and land cover change downscaling in the 21$^{st}$
century. The black and red colors represent using the top 5% and 10% parameters, respectively.

## 4. Discussion

To date, there has been only a handful of methods for downscaling projected global land use and land cover change. For example, Oskins *et al* (2016) fitted a statistical model relating coarse-scaled spatial patterns in land cover classes to finer-scaled land cover and other explaining variables. Many more studies used complex land use modeling approach (e.g., Houet *et al* 2017, Oskins *et al* 2016, Meiyappan *et al* 2014, Hurtt *et al* 2011, Souty *et al* 2012) that combines a variety of socioeconomic processes to provide global scale land use allocations. Our results demonstrated that Demeter is an effective tool for downscaling global land use and land cover change, although it adapts a relatively simpler approach. However, choices of parameter values are critically important for a simple model, since it is possible that some complicated processes are simplified to be represented by a single parameter. Although an uncalibrated Demeter can lead to noticeable errors and uncertainties in downscaled land cover areas, our results have shown the effectiveness of the calibration efforts in minimizing the downscaling errors and constraining the uncertainties.

A central purpose of our study is to making suggestions for setting up parameters for Demeter's global applications, shown as the global optimal values in Figure 2. Interestingly, we found that the parameters of intensification ratio ($r$) and selection threshold ($\tau$) strongly affected the downscaled results, while the weights of the spatial constraints and kernel radius showed small impacts on the results. This result indicates that the selected spatial constraints (soil workability and nutrient availability) and spatial autocorrelation (measured by kernel density) provide loose constrains on the land allocation in the downscaling process, therefore the users should focus more on the quality of other parameters such as $r$ and $\tau$ to which the model is more sensitive. In addition, the intensification ratio has been strictly constrained to a range close to 1.0, suggesting that the intensification of land cover, especially cropland, may be the major contributor to the global land use and land cover change, thus spatial constraints on extensification are not very effective. We also noticed that the optimal weight for soil nutrient availability for calculating the suitability indices is zero (Figure 2) and the model. A possible reason is that the soil nutrient availability has similar spatial distribution as the cropland in ESA-CCI data, thus provides little additional information in constraining the downscaling processes (Figure S10). This result suggests that the users could ignore the input of soil nutrient availability if it is not available or difficult to collect, and the quantification of the downscaling uncertainty is not required.

There has been a number of numerical methods for model calibration, such as gradient methods (Ypma, 1995), evolutionary algorithms (Ashlock, 2006), and data assimilation techniques (Kalnay, 2002). Our calibration method is relatively simpler, and the sampling steps are relatively coarse. As a result, it is possible that the calibrated parameters can be further improved with a more rigorous calibration strategy, although these biases should be small since the sampling bins are narrow and the sensitive parameters are well constrained (Figure 2). However, our method has a few advantages for this particular global land use

and land cover change downscaling model calibration problem. First, we sampled the whole parameter
space thus our Monte-Carlo downscaling experiments can well represent the parameter uncertainties.
Second, the other methods mentioned above typically adjust model parameters and run the model
iteratively to find the parameters to hit the local or global minimum cost function value (Chong and Zak,
2013), and thus can be very time consuming due to the size of the datasets and the difficulty of algorithm
parallelization. The Monte-Carlo ensemble runs of Demeter in our method can be easily parallelized and
thus is computationally efficient. Finally, the saved downscaled results from the global Monte-Carlo
downscaling experiment can be reused for regional applications. Our study provided an optimal set of
Demeter parameters. It is worth noting that these parameters are optimized to minimize the average
discrepancies between the downscaled and historically observed land cover areas at the global scale, thus
they may need to be recalibrated when Demeter is applied to a particular region. For example, the best
estimate of the intensification ratio is 1 for a global downscaling experiment, probably due to that
intensification is a more common phenomena than extensification during the past land use and land cover
change in the past two decades as recorded by the ESA-CCI data. However, this high intensification ratio
for Crop may be more realistic for the regions with long-term agricultural history (e.g., India), while it
should become lower for the United States (US) where cropland extensification rapidly happened in the
past century. We extracted the grid cells in the conterminous US (grid cells between 25° N and 50° N, and
125° W and 65° W) and India (grid cells between 7° N and 33° N, and 68° E and 98° E), and used them
together with the same method as the global calibration to determine the optimal parameters for the US
and India, which clearly showed that the intensification ratio remained 1 for India, but moved towards
lower values for the US (Figure S9). Therefore, we recommend future efforts on examining reginal
parameterization should be made for Demeter's applications at specific regional/AEZ levels. Since some
of the key parameters have clear physical definition (e.g., the intensification ratio), while the global
optimal values could be used as a starting point, it would be helpful to review the local historical land use
change to infer these parameters when applying Demeter to a specific region.

388        In addition, although the downscaled urban land use can capture most of the variability in reality, it is

clear that Demeter's performance for urban is not as good as that for other land cover types (Figure 6). On
the other hand, accurate projection of the spatial extent and pattern of urbanization is getting more
important for better understanding its environmental, ecological and socioeconomic impacts in such an
era of rapid urbanization (Georgescu et al., 2012; Jones et al., 1990; Merckx et al., 2018; Zhang et al.,
2018). Thus, a key future effort should be made for improving the downscaling accuracy of urban land
use. The relative larger errors could be either due to the limited consideration of complex urbanization
processes and the lack of specific parameterization of the urban land cover type. While incorporating
better representation of urbanization in Demeter can be more complicated, it is possible to improve the
model performance by further parameterizing the model with more historical urban data. For example,
global satellite-observed nightlights have been used for mapping urban area (Elvidge et al., 2009; Li and
Zhou, 2017b; Zhou et al., 2014) and producing a global record of annual urban dynamics (1992-2013) (Li
and Zhou, 2017a), which will be particularly useful for the future calibration of Demeter on urban
dynamics.
Model calibration usually can provide several sets of parameters to allow the calibrated model to give
similar results, which is called equifinality (Beven and Freer, 2001). As a result, the calibrated parameters
become another source of uncertainty in model-simulated results. The equifinality also exists in our
calibrations. We have observed noticeable growing uncertainties in downscaled land cover areas while
propagating the parameter uncertainties into the Demeter downscaling practices with GCAM projected
LULCC in the 21$^{st}$ century. Therefore, while calibration can remarkably reduce the uncertainty of the
parameters, it may be better to use sets of constrained parameters rather than a single set of 'best'
parameters in the practice of Demeter, for the purpose of accounting for the parameter uncertainty and
providing more reliable land use and land cover change downscaling. Moreover, it is worth noting that the
calibrated parameters are tuned for FLTs, which we believe have covered most land cover types and are
directly useful in most cases. When the users need to consider more FLTs in their global applications, the
optimal values introduced in this study can be used as a starting point for further tuning.

**5.  Conclusions**
We developed a Monte-Carlo ensemble experiment for Demeter, a land use and land cover change
downscaling model of GCAM, analyzed the model's sensitivity to its key parameters, and calibrated the
parameters to minimize the mismatch between the model-downscaled and satellite-observed land use and
land cover change in the past two decades. We identified the optimal parameter values for global
applications of Demeter, and showed that the parameterization of Demeter substantially improved the
model's performance in downscaling global land use and land cover change. The intensification ratio and
selection threshold turned out to be the most sensitive parameters, thus need to be carefully tuned,
especially when Demeter is used for regional applications. Further, the small uncertainty of parameters
after calibration can result in considerably larger uncertainties in the results when propagating them into
the practice of downscaling GCAM projections, suggesting that Demeter users consider the
parameterization equifinality to better account the uncertainties in the Demeter downscaled land use and
land cover changes.


**Code Availability**
The source code of GCAM and Demeter is available at https://github.com/JGCRI/gcam-core
and https://github.com/IMMM-SFA/demeter. The scripts for performing the calibration and analysis are
available at https://drive.google.com/open?id=1qNzh4eKgVcO_BjG2RjAw33whqxSMH8wm.

**Data Availability**
The ESA-CCI data was downloaded from https://www.esa-landcover-cci.org/. Other data are available at
https://drive.google.com/open?id=1qNzh4eKgVcO_BjG2RjAw33whqxSMH8wm.

**Author contribution**
M.C. conceived the study and all the authors contributed to design the study. M.C. lead the data
acquisition and performed the experiment and analysis with technical assistance from C.V.; M.C. wrote
the manuscript with the inputs from all the coauthors.

**Competing interests**
The authors declare that they have no conflict of interest.


**Acknowledgements**

449       This research was supported by the U.S. Department of Energy, Office of Science, as part of research

in Multi-Sector Dynamics, Earth and Environmental System Modeling Program.

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
