# Peer review of "Calibration and analysis of the uncertainty in downscaling global land use and land cover projections from GCAM using Demeter (v1.0.0)"

_Geoscientific Model Development, 2018_

## Referee Comment (RC1) · Anonymous Referee #1 · 29 Jan 2019

This study developed a method to calibrate key parameters in Demeter, a community spatial downscaling model, by using a long-term global satellite-based land cover dataset. The sensitivities of the key parameters and propagation of the uncertainties in the projection were also evaluated. The parameterization in the Demeter is important for a better performance of downscaling land use and land cover data from projection at the regional level. I'd recommend accepting this paper upon some minor revisions. 1. The section of introduction needs to be improved. The first paragraph discussed multiple topics such as background and motivations. They can be separated as individual paragraphs for readers' understanding. The challenges can also be briefly discussed in the introduction. 2. Although the Demeter has been published, an overview illustration of this model will be very helpful for readers to understand the work in this paper without reading the Demeter paper. 3. As this paper focused on the calibration of parameters, these parameters are important and deserve some explanations. For example, it is not clear what is selection threshold. 4. I read the paper in pdf. I found the symbols in the equations do not show up. They are "sum"? 5. Figure 7 can be improved regarding readability. For example, some boundaries of AEZ can be removed? 6. Figure 8 can be improved. 7. Figure 8: I expected the uncertainty will increase monotonously. But for some land cover types, it even decreases after the middle of the century. Any explanations? 8. This study is an important contribution to the development of the community spatial downscaling model, Demeter. It is still worth to discuss the limitations and future directions. For example, a set of global parameters were used in the Demeter. Further efforts should be made for the regional level and even AEZ level parameterization, especially with the capability of parallelized computing. The second effort should be made in the future work is the improvement of urban land use. Currently, the performance of urban land use is not good as other land cover types. It could be due to the limited consideration of complex urbanization process as well as the input historical urban data. More spatially and temporally consistent urban extent data can be explored (references: A global map of urban extent from nightlights; A global record of annual urban dynamics (1992-2013) from nighttime lights) in the future research. With some minor revisions, I would like to see this paper published.

---

## Short Comment (SC1) · 30 Jan 2019

Dear authors,

in my role as Executive editor of GMD, I would like to bring to your attention our Editorial version 1.1: http://www.geosci-model-dev.net/8/3487/2015/gmd-8-3487-2015.html This highlights some requirements of papers published in GMD, which is also available on the GMD website in the 'Manuscript Types' section: http://www.geoscientific-model-development.net/submission/manuscript_types.html

In particular, please note that for your paper, the following requirement has not been

met in the Discussions paper:

- "The main paper must give the model name and version number (or other unique identifier) in the title."

As I understand from the abstract, the real model development published here is demeter. Therefore please add the models name and version number to the title of your revised manuscript. E.g. "Calibration and analysis of the uncertainty in downscaling global land use and land cover projections from GCAM using DEMETER (v x.y)"

Additionally, please note, that GMD is encouraging authors to provide a persistent access to the exact version of the source code used for the model version presented in the paper. As explained in https://www.geoscientific-model-development.net/about/manuscript_types.html the preferred reference to this release is through the use of a DOI which then can be cited in the paper. For projects in GitHub (such as demeter) a DOI for a released code version can easily be created using Zenodo, see https://guides.github.com/activities/citable-code/ for details.

Yours, Astrid Kerkweg

---

## Referee Comment (RC2) · Anonymous Referee #2 · 23 Feb 2019

General Comments: The paper presented a thorough investigation of the sensitivity of a global land cover / land use downscaling model (Demeter) to its six model parameters. The work provides essential foundation for making better use of Demeter, and is worth publishing. From a user's point of view, I would appreciate the following suggested modifications made, which collectively ask the authors to further interpret their experiment results and form actionable suggestions for Demeter applications.

Specific Comments: 1. The authors argued "equifinality" is present with multi-parameter models like Demeter, and presented the optimal setting and top 5% performance setting. It would be useful to see the top few (say, 5-10) "equifinality" parameter

settings. 2. The manuscript showed attempts to make suggestions for general users of Demeter, but they need to be more specific and explicit to be useful. For example, the ideal weight for soil nutrient is 0 (figure 2), meaning the model is better off without considering this input variable. Then, the implication for Demeter users is that, users don't need to worry if they don't have good input for this variable, and should focus on getting better quality input for variables that the model is more sensitive to. 3. Regional applications of Demeter: The authors stated that different regions differ from the global "average" situation in their own ways, and regional applications of Demeter require "careful" tuning, but provided no further suggestions. Although this paper focuses on a global application of Demeter and global applications are different from regional ones, the authors have learned much about the model's sensitivity, and are better positioned than any user out there to infer what are good starting points (e.g. a range of values to try first, proper sizes of increments when changing values of specific variables) when parameterizing Demeter for regional uses. This doesn't need to be long, but given the authors' knowledge about the topic, even some speculations would be helpful, but they need to be actionable. 4. Global applications: The authors presented the optimal set of parameters for global applications, then made some vague suggestions for (global) modeling tuning. Since the authors' experiment is global, it seems the optimal parameters have been identified for global applications. In what cases would global tuning be needed? And what are good starting points for such tuning? 5. Demeter's residuals show very strong spatial patterns / biases (figure 7). Some explanations about why it occurred and how it may be moderated (if possible) would be useful.

Technical Corrections: 1. The authors mentioned how Demeter compare to other spatial downscaling models, but it came up in the method section. It would be nice to see that in introduction. 2. Many equations are not displayed properly in my copy of the manuscript. Equation (5) especially is not readable at all. 3. Table 1 showing land cover conversion priorities must label whether rows/cols are origin/destination land cover types, because the conversion priorities are not symmetric. 4. Ln 152: "Y is the model outputs (i.e., E)" and the following equation E(Y|X) are confusing. Usually,

[Figure]

E(Y) denotes the statistical expectation of Y.

---

## Author Comment (AC1) · 26 Mar 2019

This study developed a method to calibrate key parameters in Demeter, a community spatial downscaling model, by using a long-term global satellite-based land cover dataset. The sensitivities of the key parameters and propagation of the uncertainties in the projection were also evaluated. The parameterization in the Demeter is important for a better performance of downscaling land use and land cover data from projection at the regional level. I'd recommend accepting this paper upon some minor revisions.

- We thank the reviewer's positive comments on the importance of this paper. Below we respond the reviewer's specific comments point-by-point.

[Figure]

1. The section of introduction needs to be improved. The first paragraph discussed multiple topics such as background and motivations. They can be separated as individual paragraphs for readers' understanding. The challenges can also be briefly discussed in the introduction.

- We have separated the second paragraph in the original manuscript into two paragraphs. In the first paragraph of the revised "introduction" section, we introduced the critical role of LULCC in the Earth system science research, followed by the statement of motivation of studying spatial downscaling of LULCC by the integrated Human-Earth system models such as GCAM. In the third paragraph, we added sentences for introducing other spatial disaggregation models as suggested by RC2, and briefly discussed the challenges of determining Demeter parameters. See Line 55-77.

2. Although the Demeter has been published, an overview illustration of this model will be very helpful for readers to understand the work in this paper without reading the Demeter paper.

- We added a figure to provide an overview of Demeter's key processes. Please see Figure S1 in the supplementary materials.

3. As this paper focused on the calibration of parameters, these parameters are important and deserve some explanations. For example, it is not clear what is selection threshold.

- We added further clarifications to the parameters and associated variables. Please see Line 105-106 and Line 115-116.

4. I read the paper in pdf. I found the symbols in the equations do not show up. They are "sum"?

- We have made modifications to the equations to make sure they show up correctly.

5. Figure 7 can be improved regarding readability. For example, some boundaries of AEZ can be removed?

- We improved the quality of Figure 7 and the related Figure S2-S6 in the supplementary file by reducing the visibility of the AEZ boundaries.

6. Figure 8 can be improved. - We have improved the quality of Figure 8 and made modifications with the results of using "top 10%" parameters as suggested by reviewer 2.

7. Figure 8: I expected the uncertainty will increase monotonously. But for some land cover types, it even decreases after the middle of the century. Any explanations?

- We thank the reviewer identify this problem. We found a mistake in preparing Figure 8. All the uncertainties increase monotonously after the correction.

8. This study is an important contribution to the development of the community spatial downscaling model, Demeter. It is still worth to discuss the limitations and future directions. For example, a set of global parameters were used in the Demeter. Further efforts should be made for the regional level and even AEZ level parameterization, especially with the capability of parallelized computing. The second effort should be made in the future work is the improvement of urban land use. Currently, the performance of urban land use is not good as other land cover types. It could be due to the limited consideration of complex urbanization process as well as the input historical urban data. More spatially and temporally consistent urban extent data can be explored (references: A global map of urban extent from nightlights; A global record of annual urban dynamics (1992–2013) from nighttime lights) in the future research. With some minor revisions, I would like to see this paper published.

- We thank the reviewer's valuable suggestions. We have inserted a paragraph of discussion on the limitations of current version of Demeter and its parameterization, and pointed out future study directions such as regional/AEZ-level parameterization and improving urban parameterization with satellite-derived urban records. Please see Lines 383-401.

Please also note the supplement to this comment:
https://www.geosci-model-dev-discuss.net/gmd-2018-248/gmd-2018-248-AC1-supplement.zip

————————————————————

---

## Author Comment (AC2) · 26 Mar 2019

Dear authors, in my role as Executive editor of GMD, I would like to bring to your attention our Editorial version 1.1: http://www.geosci-model-dev.net/8/3487/2015/gmd-8-3487-2015.html This highlights some requirements of papers published in GMD, which is also available on the GMD website in the 'Manuscript Types' section: http://www.geoscientific-modeldevelopment.net/submission/manuscript_types.html In particular, please note that for your paper, the following requirement has not been met in the Discussions paper: "The main paper must give the model name and version number (or other unique identifier) in the title." As I understand from the abstract, the

real model development published here is demeter. Therefore please add the models name and version number to the title of your revised manuscript. E.g. "Calibration and analysis of the uncertainty in downscaling global land use and land cover projections from GCAM using DEMETER (v x.y)".

- We have updated the title as "Calibration and analysis of the uncertainty in downscaling global land use and land cover projections from GCAM using Demeter (v1.0.0)"

Additionally, please note, that GMD is encouraging authors to provide a persistent access to the exact version of the source code used for the model version presented in the paper. As explained in https://www.geoscientific-modeldevelopment.net/about/manuscript_types.html the preferred reference to this release is through the use of a DOI which then can be cited in the paper. For projects in GitHub (such as demeter) a DOI for a released code version can easily be created using Zenodo, see https://guides.github.com/activities/citable-code/ for details.

- We have added the DOI in the revised manuscript. See Line 60.

Please also note the supplement to this comment:
https://www.geosci-model-dev-discuss.net/gmd-2018-248/gmd-2018-248-AC2-supplement.zip

---

## Author Comment (AC3) · 26 Mar 2019

Please see our responses and revised manuscript in the supplements.

Please also note the supplement to this comment:
https://www.geosci-model-dev-discuss.net/gmd-2018-248/gmd-2018-248-AC3-supplement.zip